# Spatiotemporal Distribution of CO in the UTLS Region in the Asian Summer Monsoon Season: Analysis of MLS Observations and CMIP6 Simulations

**Ziling Liang [1,2], Fangrui Zhu [1], Tian Liang [1], Fuhai Luo [1] and Jiali Luo [1,*]**

[1]  Key Laboratory for Semi-Arid Climate Change of the Ministry of Education, College of Atmospheric Sciences, Lanzhou University, Lanzhou 730000, China
[2]  Department of Atmospheric and Oceanic Sciences, McGill University, Montreal, QC H3A 0B9, Canada
[*]  Correspondence: luojl@lzu.edu.cn

**Abstract:** In this study, CO is used as a tracer to evaluate the chemical field related to the Asian summer monsoon anticyclone (ASMA) in the upper troposphere and lower stratosphere (UTLS) region simulated by Coupled Model Intercomparison Project Phase 6 (CMIP6) climate models from a multi-spatiotemporal perspective. The results show that the simulations of the six selected CMIP6 global climate models are well correlated with the MLS observations, while each model has its own advantages and disadvantages in the simulation of the ASMA and related chemical and geopotential height fields. Compared with MLS data, all six CMIP6 models can reasonably simulate the high CO values and the corresponding anticyclone, although certain biases exist in the simulations. Each model output has certain degrees of deviation in the simulation of the ASMA center position. In terms of time series, the six CMIP6 global models all exhibit an interannual variation CO mixing ratio over the ASM region while the interannual variation features are different from that in MLS. In general, it is impossible to identify a single determined model that can well reproduce the observations. In future work to assess the development trend and location of the ASMA, simulations of CESM2-WACCM and GFDL-ESM4 might be used due to their better performance than other models.

**Keywords:** Asian summer monsoon; CMIP6; upper troposphere and lower stratosphere; CO

## 1. Introduction

The Asian summer monsoon (ASM) is one of the most important components of the global climate system [1]. During the ASM season, the Asian summer monsoon anticyclone (ASMA) occupies the upper troposphere and lower stratosphere (UTLS) from June and disappears at the end of August. It is the most important atmospheric circulation in the UTLS of the Northern Hemisphere in summer. Extensive studies have been conducted in recent years to investigate the dynamical characteristics of the ASMA circulation and its influence on the distribution of atmospheric chemical components [2–5].

It is found that a westerly jet is located to the north of the ASMA and an easterly jet is located to the south [2,4]. Such a circulation configuration makes it easy for the anticyclone to trap air inside the ASMA [1]. The frequent deep convective activities in the southeast of the ASM and the siege effect of the anticyclonic circulation promote rapid dispersion of insoluble pollutants and aerosols from near ground level to the UTLS region, from which they can enter the stratosphere and are rapidly transported globally [2]. As a result, anomalous distribution of atmospheric chemistry can be found within the ASMA, which is reflected in significant increases in the concentrations of tracers such as CO, $H_2O$, HCN and a large number of hydrocarbons in the boundary layer and troposphere. Meanwhile, as a stratospheric tracer, the concentration of $O_3$ has decreased significantly [3,4]. Through stratosphere-troposphere exchange (STE), this type of pollutant dispersion may have an impact on local or even global atmospheric environment [5]. Therefore, it is of great

scientific importance to fully understand the distribution of chemical tracers in the UTLS region during the ASM season. Furthermore, pollution emissions in Asia have increased dramatically following the rapid population growth and continuous economic development in recent decades [6]. It is also necessary to investigate how the pollution dispersion caused by the ASM circulation affects the global atmospheric environment, human production and life in the future.

CO is widely used as a trace gas in ASMA-related STE studies since it is a chemical component emitted near-surface and has a photochemical lifetime of about 2 months in the troposphere [7,8]. Previous studies have found that there is an obvious high CO concentration center in the UTLS region, which well corresponds with the ASMA center during the ASM season [1,9]. Therefore, CO can be used to characterize the location of the ASMA center and its spatiotemporal variation. However, note that the ASMA itself shows significant interannual variability [10,11]. Further research is required to determine if the distribution of chemical components such as CO also exhibits interannual fluctuation. If so, what are the variation features?

Satellite observations provide a reliable source of information for the ASMA studies in the UTLS region. The Microwave Limb Sounder (MLS), an active remote sensing satellite Aura in Sun-synchronous orbit, can provide systematic and spatiotemporally continuous high vertical resolution CO information with the spatial coverage up to the whole Earth [12]. However, its temporal coverage is limited. Climate models provide an effective tool for the climate-chemistry interaction study and climate simulation as well as the prediction of future climate change [13]. Zhou and Yu pointed out that climate models can well reproduce the primary characteristics of the ASMA under natural and anthropogenic forcing scenarios, and climate model simulations are important for the study of climate change mechanisms and climate change attribution [14]. The Coupled Model Intercomparison Project Phase 6 (CMIP6) was initiated in 2014 to meet the increasing scientific demands of the broad climate science community and address new challenges emerging in climate modeling. The analysis of CMIP6 simulations will promote our understanding of the most pressing problems in climate variability and change study [15]. Several previous studies have been conducted to evaluate simulations of temperature and precipitation during the ASM period from multiple models participating in the CMIP5 and CMIP6 [16]. Thus, to evaluate temporal and spatial distributions of chemical tracers such as CO during the ASM season is of great significance.

In this study, we use CO data extracted from the MLS dataset and the NCEP reanalysis product to evaluate the capability of six CMIP6 models for the simulation of the ASMA in the UTLS layer from a multi-temporal perspective. This study also aims to provide a reliable scientific basis for the improvement of global climate models and the prediction of future ASMA evolution. This paper is organized as follows. Section 2 describes the satellite data, meteorological data and six CMIP6 models used in the paper. Section 3 displays the methodology. Section 4 evaluates the model performance with satellite and reanalysis data. Section 5 provides a brief summary of the conclusion.

## 2. Data Description

### 2.1. Satellite Data

The MLS is a microwave limb sounder on board the polar-orbiting satellite Aura with a spatial coverage of 82°N–82°S and 180°E–180°W. The horizontal resolution is 4.5 km × 450 km at 100 hPa with an accuracy of 14 ppb [17], and the CO information used in this paper is obtained from radiance measurements at two bands in the MLS 240 GHz radiometer [18]. In this paper, monthly average data of MLS V004 L3 CO from 2005 to 2014 with a horizontal resolution of 4° × 5°, a vertical resolution of 3–6 km and an effective altitude of 215–0.0046 hPa are used.

### 2.2. Meteorological Analysis Data

The National Centers for Environmental Prediction (NCEP) reanalysis data of horizontal wind, geopotential height, the tropopause height and potential temperature are used to demonstrate spatiotemporal distribution of the ASMA and its correspondence with satellite observation data. The time span is from June to August over the years 2005–2014. These averaged data at monthly time steps are on global $1° \times 1°$ grids at 26 barometric (1000 hPa to 10 hPa) levels [19].

### 2.3. CMIP6 Models

In this study, we evaluate CO mixing ratio in the UTLS derived from six models that participate in CMIP6: BCC-ESM1, CESM2-WACCM, CESM2-WACCM-FV2, EC-Earth3-AerChem, GFDL-ESM4, MRI-ESM2-0.

BCC-ESM1 is a new version of the Earth System Model (ESM) developed by the Beijing Climate Center (BCC), which includes interactive atmospheric chemistry and aerosols. The atmospheric component of BCC-ESM1 is BCCAGCM3-Chem [20]. Community Earth System Model Version 2—the whole atmosphere community climate model (CESM2-WACCM) is a comprehensive Earth system model with coupled atmosphere, land, ocean, sea ice and glacier, which is jointly developed by scientists, software engineers and students from the National Center for Atmospheric Research (NCAR) and various universities and research institutes. The atmospheric component of CESM2 is CAM6 [21,22]. Improvements of CESM2-WACCM compared to CESM-WACCM include adjustments of atmospheric physics parameterization schemes, many new capabilities in the middle and upper atmosphere and improvements of the chemical modules, etc. EC-Earth3-AerChem is basically a global climate and earth system model, which is an extension of EC-Earth3 with an additional component to simulate aerosols and atmospheric chemistry. It is developed by the European consortium of meteorological services, research institutes, and high-performance computing centers [23]. The Earth System Model Version 4.1 (ESM4.1) of the Geophysical Fluid Dynamics Laboratory Climate Model (GFDL) is based on components and coupled model developments at GFDL over 2013–2018, when GFDL has doubled the horizontal resolution of both atmosphere and ocean and contributed to CMIP6 development [24]. Meteorological Research Institute Earth System Model version 2.0 (MRI-ESM2.0) has been built on the basis of MRI-CGCM3 and MRI-ESM. The improvements of various cloud schemes in these models significantly reduce radiation errors at the top of the atmosphere compared to those in the CMIP5 models [25].

We use the results of CMIP6 historical simulations to evaluate CO distribution in the UTLS during the ASM season. It spans the period from 1850 to 2014 when extensive instrumental temperature measurements are available. Under this condition, model outputs can be evaluated against the present climate and observed climate change. The Community Emission Data System (CEDS) provides the consistent emission inventory of CO for CMIP6 [26]. We select results from 2005 to 2014 to compare with MLS observations. To analyze future trajectories and development of the ASMA, we investigate changes in CO across a series of scenarios (shared socioeconomic pathways; SSPs) developed for The Scenario Model Intercomparison Project (ScenarioMIP) [27]. SSP1 describes the optimistic trends for human development with strong economic growth via sustainable pathways. SSP5 describes a world of rapid economic development at the cost of the dramatic effects of climate change. In both SSP1 and SSP5 scenarios, the income of the residents has increased substantially, and the lack of food has been greatly reduced. The biggest difference is that SSP5 requests huge consumption of fossil fuels. SSP2 is a middle-of-the-road scenario in which the trends remain their historical patterns with moderate population and economic growth. SSP3 depicts more pessimistic development trends with regional security as the priority. In order to evaluate a variety of potential possibilities, we select outcomes under four SSPs with specific ranges of forcing: SSP1-1.9 ($+1.9$ W m$^{-2}$), SSP1-2.6 ($+2.6$ W m$^{-2}$), SSP2-4.5 ($+4.5$ W m$^{-2}$), SSP3-7.0 ($+7.0$ W m$^{-2}$), and SSP5-8.5 ($+8.5$ m$^{-2}$).

Climate models are a set of mathematical and physical equations describing the climate system established by a series of fundamental physical and chemical laws. To compare satellite observations with model simulations, the historical simulations of the models after the industrial revolution are used. Monthly mean simulations of models with complete available dynamic fields and CO fields from 2005 to 2014 are selected for analysis. Satellite observations and model outputs overlap during the period 2005–2014. The information of the 6 models is listed in Table 1.

**Table 1.** Summary information of six CMIP6 global climate models.

| Model | Country | Organization | Resolution |
|---|---|---|---|
| BCC-ESM1 | China | BCC | $2.8° \times 2.8°$ |
| CESM2-WACCM | America | NCAR | $0.94° \times 1.25°$ |
| CESM2-WACCM-FV2 | America | NCAR | $1.9° \times 2.5°$ |
| EC-Earth3-AerChem | Europe | EC-Earth-Consortium | $2° \times 3°$ |
| GFDL-ESM4 | America | NOAA-GFDL | $2° \times 3°$ |
| MRI-ESM2-0 | Japan | MRI | $2.8° \times 2.8°$ |

## 3. Method

Due to the mismatched resolution between individual model outputs and between model outputs and satellite data and meteorological analysis data, the bilinear interpolation method is used to remap all the data to $3° \times 3°$ grids for quantitative comparison. 100 hPa is then chosen as the pressure level at which the temporal fluctuation of CO is examined. The area over 0°–60°N and 0°–150°E is selected as the larger horizontal area for comparison, while major analysis is conducted in the area of ASM over 10°–40°N and 30°–130°E [28,29].

In addition, because the model simulations do not provide thermal tropopause height, the definition of the tropopause provided by the World Meteorological Organization is adopted in the present study, i.e., the lowest altitude where the rate of temperature is reduced to 2 K km$^{-1}$ or below and the average temperature decrement rate in the atmosphere within 2 km of this altitude is less than 2 K km$^{-1}$ is determined to be the tropopause height.

## 4. Results

Figure 1 shows the horizontal distribution of seasonal mean CO mixing ratio at 100 hPa over the ASM region from the six CMIP6 simulations and MLS observations during June–August (JJA) for the period 2005–2014. The geopotential height contours and horizontal wind fields indicate the seasonal mean position of the ASMA.

All six models produce an evident anticyclone circulation and a relatively high CO center at 100 hPa. That is to say, all these models are able to capture the ASMA and its confinement features. The CO mixing ratio within the ASMA is about 60–75 ppbv in MLS but model outputs show different CO values. Specifically, the CO mixing ratio and the center of relatively high CO from CESM2-WACCM are similar to MLS observations, while higher values of CO mixing ratio are found in the simulations of GFDL-ESM4, BCC-ESM1 and EC-Earth3-AerChem. An area of abnormally high CO value can also be found to the southwest of the ASMA in the simulation of EC-Earth3-AerChem. The geopotential height fields simulated by CESM2-WACCM-FV2 is similar to that in the NCEP reanalysis but the simulated CO mixing ratio is lower than MLS observations. Although a relatively high CO center can be found in the simulation of MRI-ESM2-0, the simulated GPH and CO centers both show a slight displacement to the southeast of their observations. Among all, the geopotential height fields simulated by GFDL-ESM4, BCC-ESM1, EC-Earth3-AerChem and MRI-ESM2-0 are relatively low. CESM2-WACCM simulations of the geopotential height field are relatively high and its simulation of CO mixing ratio is more consistent with MLS observations. Overall, the CESM2-WACCM simulations of geopotential fields and CO agree better with MLS observations than the simulations from other models.

ASMA is a three-dimensional system connected with persistent tropospheric upward motions [12,30]. The relatively high concentration of CO in the UTLS region is attributed to tropospheric transport and anticyclonic confinement [30]. Park et al. pointed out that the vertical structure of the ASMA may help explain the transport of atmospheric constituents to the tropopause level [12]. Thus, we further compare the vertical distribution of CO over the ASM region in JJA.

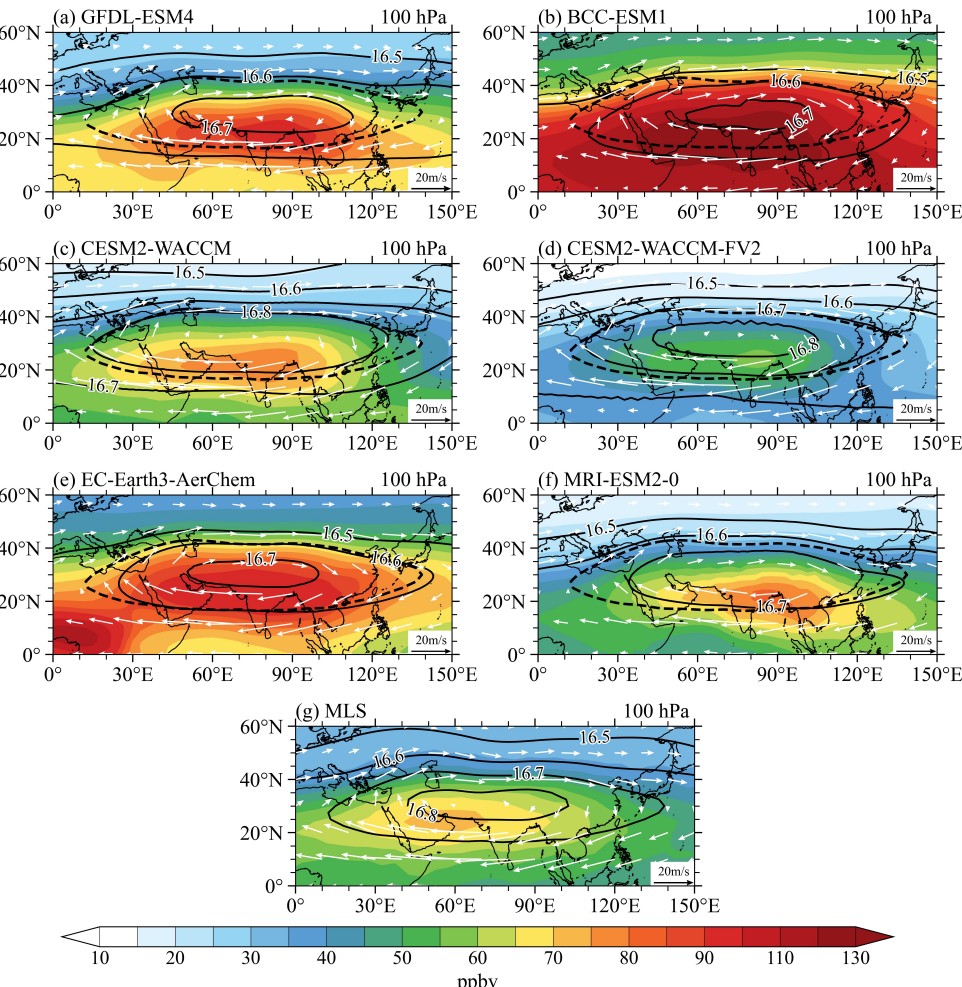

**Figure 1.** Climatology of CO mixing ratio (color shaded, ppbv) at 100 hPa for the simulations of six CMIP6 models and MLS observations during JJA 2005–2014. Black contours represent the geopotential height (GPH) from each model (**a–f**) and NCEP data (**g**). The white vectors indicate wind fields from models (**a–f**) and NCEP data (**g**). The black dashed contour in (**a–f**) is the 16.7 gpm contour of GPH from NCEP data.

Figure 2 displays height-latitude profiles of seasonal mean CO between 0°–60°N from the CMIP6 model simulations and MLS observations (JJA, 2005–2014). We also show the potential temperature and thermal tropopause in the figures and they are similar in the reanalysis and model simulations. Most of the models simulate the arch shape of the MLS chemical fields in the upper troposphere, which is the most evident feature shown in Figure 2. The enhanced CO mixing ratio throughout the whole troposphere with a relatively high CO center located in the upper troposphere between 10°–40°N. Enhanced CO can also be found above the tropopause within the ASMA region, and this corresponds to the maximum CO observed over the ASMA shown in Figure 1. In the simulation of BCC-ESM1, in addition to the CO enhancements between 10°–40°N shown in other models, the relatively high CO region near the tropopause expands towards the tropics. There exist

upward motions in the enhanced CO region in both the reanalysis and the simulations of BCC-ESM1, CESM2-WACCM, CESM2-WACCM-FV2 and EC-Earth3-AerChem, while upward motions in other models are weak. Overall, all the six CMIP6 models simulate a high CO in the troposphere at altitudes near the tropopause within the area of anticyclone.

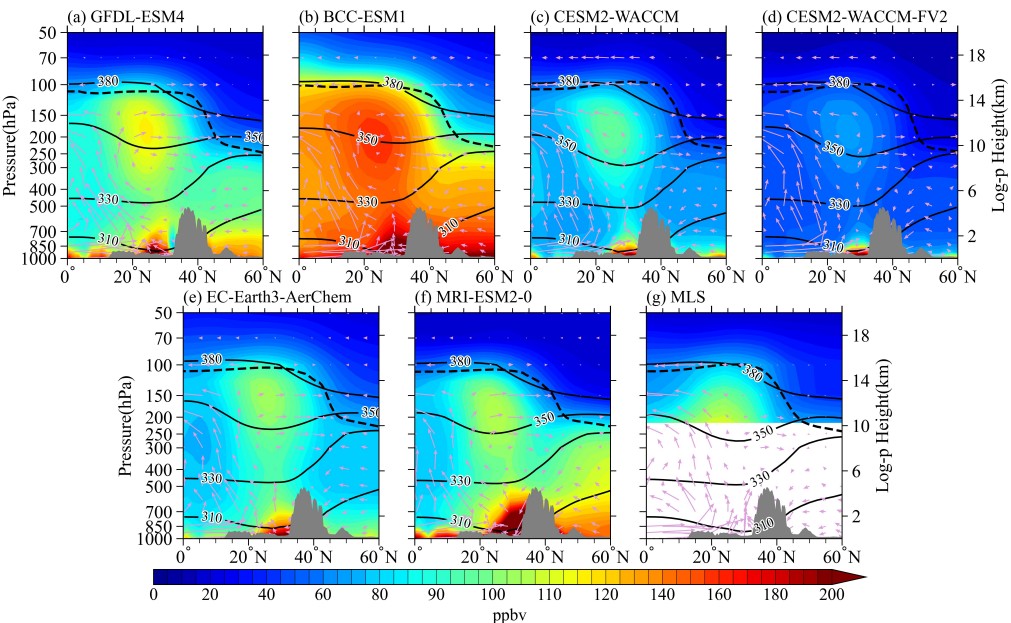

**Figure 2.** Latitude-height cross sections of JJA mean CO mixing ratio (color shaded, ppbv) between 30°–130°E during 2005–2014 JJA. Thin black solid lines are potential temperature. Thick black dashed lines represent the thermal tropopause. Purple arrows indicate the wind fields. Grey shadings indicate topography. (**a–f**) are from models and (**g**) is from MLS and NCEP reanalysis.

Figure 3 further shows longitudinal CO vertical distribution and dynamical factors. In spite of MLS, CO is only available above 200 hPa, a significant and relatively high CO center is found over the Tibetan Plateau. It is seen that although the values of the CO mixing ratio from different models are different, they all show an enhancement feature over the Tibetan Plateau. Note that the relatively high CO centers are associated with upward motions below. Tropospheric CO from BCC-ESM1 is abnormally high compared to MLS CO while that from CESM-WACCM-FV2 is abnormally low. In addition, there are also relatively high CO regions over 0°–20°E in GFDL-ESM4, EC-Earth3-AerChem, and MRI-ESM2-0. From the perspectives of CO mixing ratio, CO vertical distribution and dynamical fields, results from CESM2-WACCM are the closest to observation.

Although relatively high CO center in the UTLS can be observed in MLS and model outputs, important distribution differences exist between them (Figures 1–3). To further understand the spatial distribution of CO at 100 hPa and the results in CMIP6 models, we analyze the distribution characteristics of maximum CO (designated as the CO center) over the ASM region in JJA (Figure 4). Based on MLS observations, the CO maximum in JJA is around 65–90 ppbv and the peak value occurs at about 50°E. CO from GFDL-ESM4 has a similar mixing ratio in this region but the peak value occurs at about 90°E. CEMS2-WACCM shows a relatively lower CO concentration, and the maximum CO is also found to the east of that shown in MLS. Same as the results shown in Figure 1, CO from BCC-ESM1 and EC-Earthe3-AerChem is higher than that from MLS, whereas the results from CESM2-WACCM-FV2 are lower. At about 10°E, there exists another peak of CO in the simulation of EC-Earth3-AerChem, which corresponds to the abnormally high CO area to the southwest of the ASMA. MRI-ESM2-0 yields lower results in June. Note that CO from MLS always peaks at about 50°–60°E, whereas all the model simulations show a shift to the east of the MLS high CO center. It is seen from Figure 3 that strong upward motions in models are over about 90°E which is consistent with the regions of CO peak in Figure 4.

The location of the upward motions and the relatively coarser horizontal resolution of MLS compared to that of the models may possibly be responsible for the discrepancy.

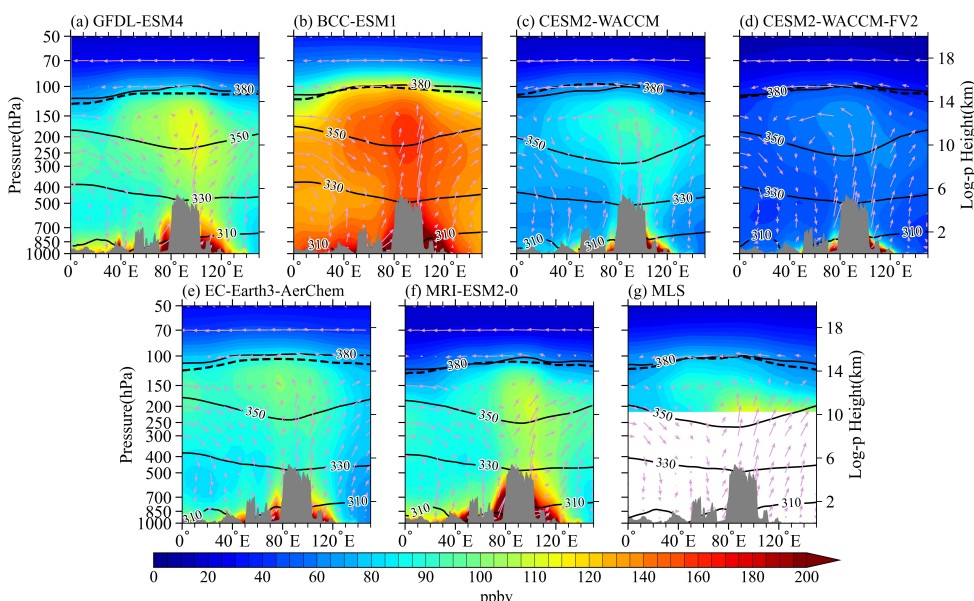

**Figure 3.** Same as Figure 2, but for longitude-height cross sections between 10° and 40°N.

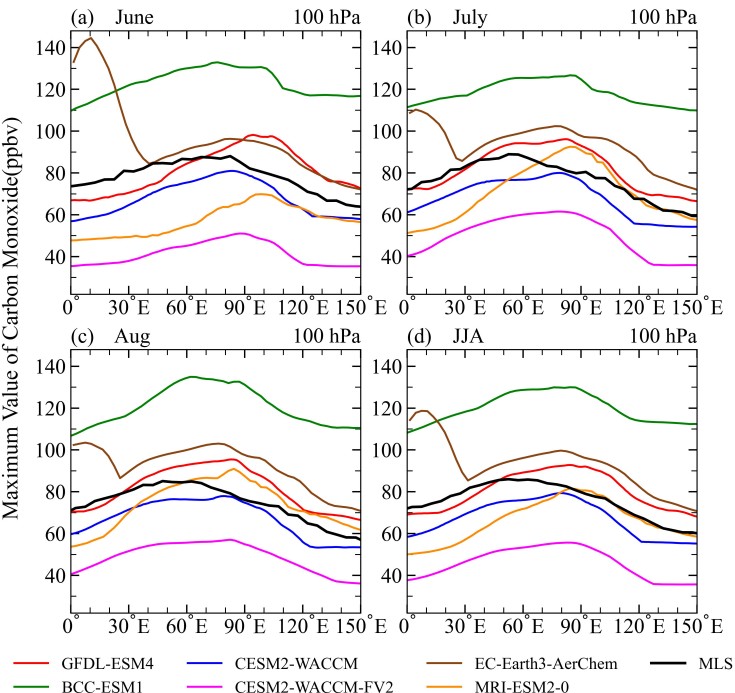

**Figure 4.** Maximum CO values between 0°–60°N for JJA seasonal mean and monthly means of June, July and August from CMIP6 models and MLS observations at 100 hPa during 2005–2014 (Unit: ppbv).

Previous research indicated that the ASMA varies in intensity and location year by year [1,12]. Concentrations of tropospheric tracers in the UTLS during the ASM season are influenced by surface emissions and the ASMA intensity [1,12,28,30]. To further explore the interannual variation of CO and evaluate simulations of the six CMIP6 models, Figure 5 shows 30°–130°E mean CO mixing ratio at 100 hPa during the ASM season over 2005–2014. MLS observations reveal an evident south–north oscillation of relatively high CO center

and interannual variation of CO mixing ratio. Specifically, the CO mixing ratio in 2009, 2012 and 2014 is greater than 80 ppbv and higher than that in other years. It is also higher than the multi-year mean value (Figure 1g). The CO mixing ratio is the lowest in 2013. Meanwhile, the relatively high CO center was located to the north of 20°N and expanded to about 35°N in 2009. Despite the fact that the CO mixing ratio in 2012 and 2014 is greater than that in other years, the relatively high CO center in these two years was located to the south of 30°N and expanded to near 15°N. The CO mixing ratio in the simulations of the six models all exhibits an annual variation, but its south–north oscillation and annual variation features are different from that in MLS. We can also find an east–west oscillation of the relatively high CO region from MLS (Figure 6). In the years 2009, 2012, and 2014, the relatively high CO region is located over the area from 30°E to 110°E based on MLS data. However, this feature is not shown in model outputs. From Figures 5 and 6, we can find that the CMIP6 models cannot reproduce the CO interannual variation over the ASM region.

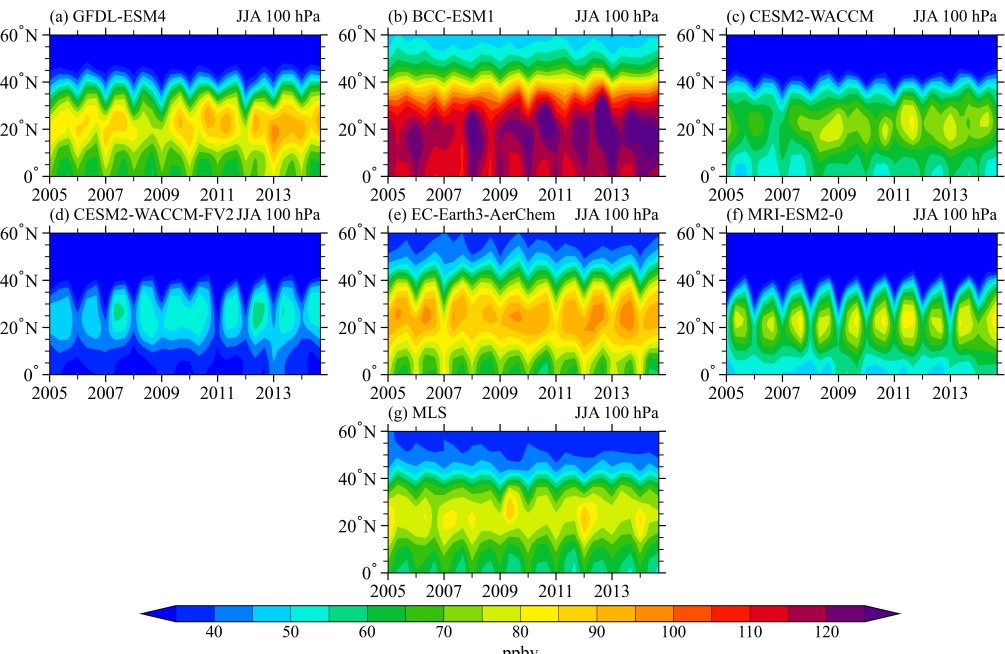

**Figure 5.** Latitude-time variations of CO mixing ratio (ppbv) at 100 hPa from models (**a**–**f**) and MLS data (**g**). CO mixing ratio is averaged over 30°–130°E.

Figure 7 displays the spatial correlation coefficients and the root mean square errors of 2005–2014 mean CO mixing ratio in the six model simulations compared to the observations over the ASM region in June, July, August and the summer mean (JJA). Each point corresponds to a single simulation of a specific model. The radial distance of the point from the origin is the ratio of the standard deviation of the model simulated CO with respect to MLS CO. The azimuthal location of the point indicates the pattern correlation coefficient between the simulated CO and MLS CO. The observation point (the reference point for comparison with model simulations) is on the abscissa with one unit of standard deviation.

The spatial correlation coefficient varies between 0.8 and 0.96 for the CMIP6 models, indicating that the simulated CO distribution at 100 hPa during the ASM season matches well with MLS observations. The results in July and August are better than that in June. The root mean square error ranges between 10 and 40. GFDL-ESM4 results have the smallest ratio. CESM2-WACCM and EC-Earth3-AerChem results also have small ratios. However, we have demonstrated that an abnormally high CO region is located to the southwest of the ASMA in EC-Earth3-AerChem (Figure 1).

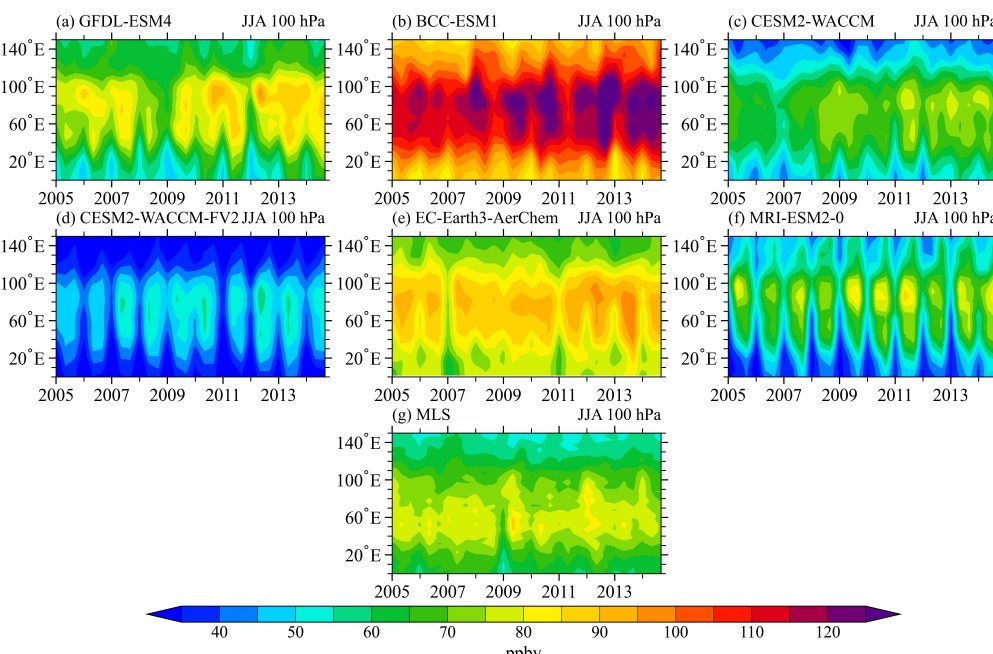

**Figure 6.** Same as Figure 5, but for CO mixing ratio averaged over 10°–40°N.

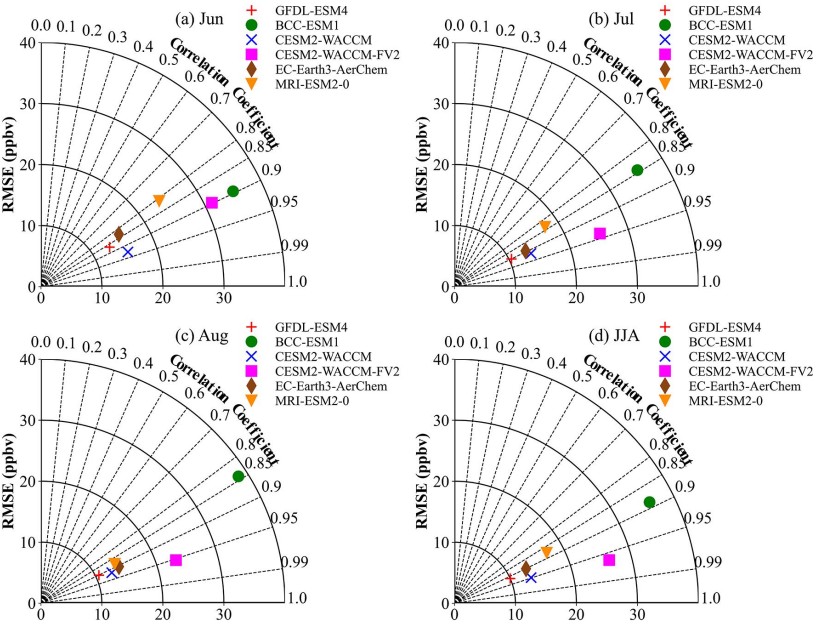

**Figure 7.** Spatial correlation coefficients and root mean square errors of the 100 hPa CO mixing ratio from CMPI6 model simulations with MLS observations for June (**a**), July (**b**), August (**c**), and JJA mean (**d**) during 2005–2014. Angular axes show correlations between 100 hPa CO mixing ratio from MLS and individual model simulation; radial axes show standard deviation (root-mean-square deviation, unit: ppbv). Each symbol represents a model.

Based on the above analysis, we use the results of CESM2-WACCM and GFDL-ESM4 to investigate projected changes in JJA mean CO mixing ratio at 100 hPa over the ASM region under different scenarios. It is found that the CO mixing ratio at 100 hPa will increase to about 2 times larger than its 2005–2014 mean in 2100 under the SSP 3-7.0 scenario (Figures 1 and 8), but decrease under the SSP 1-1.9, SSP 1-2.6 and SSP 2-4.5 scenarios. The CO mixing ratio over the ASM region under the SSP 5-8.5 will also increase to larger than 100 ppbv at 2100. That is to say, negative CO trends are found under the SSP 1-1.9, SSP

1-2.6 and SSP 2-4.5 scenarios while positive trends are shown under the SSP 5-8.5 and SSP 3-7.0 scenarios. Figure 9 further shows that the CO mixing ratios at 100 hPa under the SSP 2-4.5 and SSP 5-8.5 are similar before 2050. By the end of the 21st century, the CO mixing ratio shows consistent decreasing trajectories under the two SSP1 scenarios. After 2050, the CO mixing ratio at 100 hPa under the SSP 5-8.5 will increase but it will decrease under the SSP 2-4.5. The trajectories under the SSP1 and SSP2 both indicate a transition towards less polluting [27,31]. With the implementation of more sustainable practices, the SSP1 gives a vision of rapid economic growth and a clean future. In the SSP2-4.5, a continuation of the historical model shows significant negative emissions after the middle of the century. Despite the optimistic projections designed for the future, the SSP5-8.5 has always been dependent on fossil-fuel drivers, and emissions keep an upward trend after 2050. In the SSP3-7.0 world, less investment in technology and faster growth in population lead to higher pollutant emissions shown in the trajectories. In addition, the positive trend under the SSP 3-7.0 in the simulation of CESM2-WACCM is larger than that from GFDL-ESM4, describing a more pessimistic future.

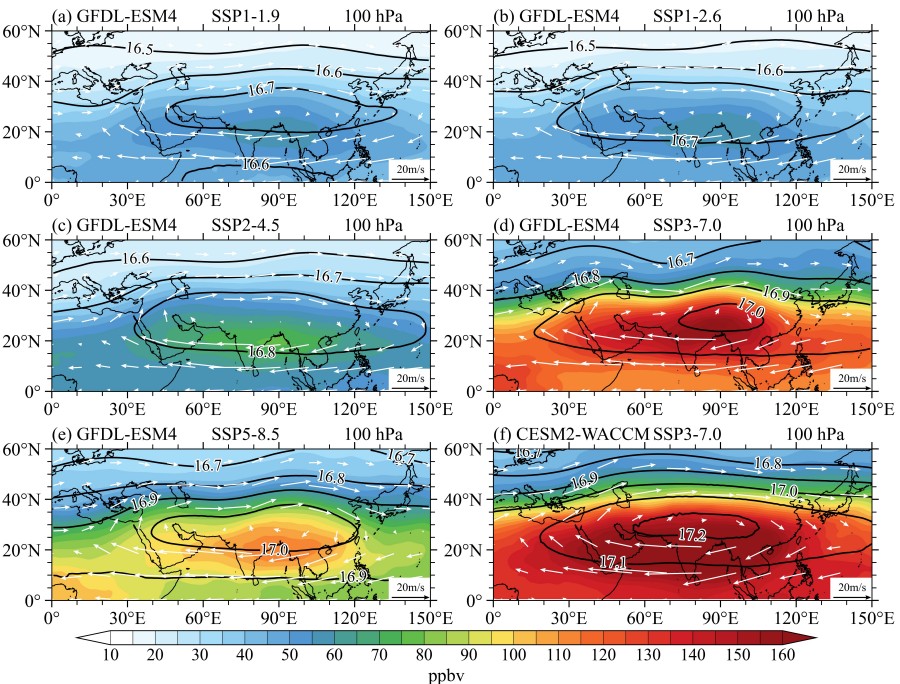

**Figure 8.** Climatology of CO mixing ratio (color shaded, ppbv) at 100 hPa from simulations of GFDL-ESM4 and CESM2-WACCM for JJA 2100 under different scenarios. Black contours represent geopotential height (GPH) from each model (**a**–**f**). The white vectors indicate wind field from models (**a**–**f**).

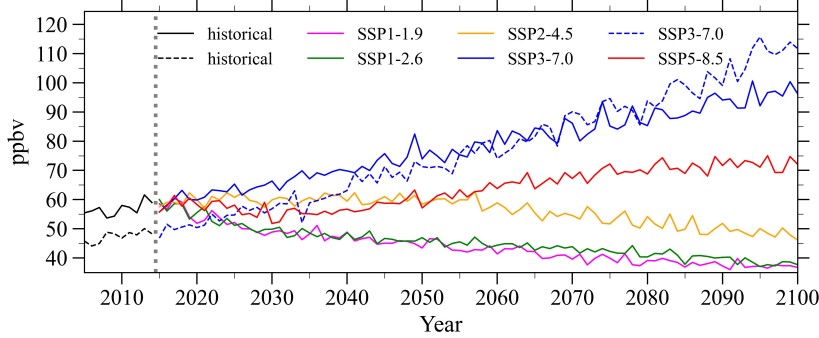

**Figure 9.** Time series of JJA mean CO mixing ratio (ppbv) within the ASM region (0°–60°N, 0°–150°E) at 100 hPa under different scenarios. Solid lines indicate results from GFDL-ESM4 and dashed lines are results from CESM2-WACCM.

## 5. Summary

This study investigates the spatial structure and interannual variation of the Asian summer monsoon anticyclone (ASMA), which is the dominant circulation feature in the UTLS region above Asia. A series evaluation of six global climate models in the Coupled Model Intercomparison Project Phase 6 (CMIP6) assisted by observations from the MLS satellite dataset as well as the NCEP reanalysis product is conducted in this study.

The capability of the CMIP6 models to reproduce the ASMA and its confinement features is satisfactory, yet the model simulations of the CO mixing ratio are less accurate than their simulations of the ASMA. CESM2-WACCM yields values of CO mixing ratio similar to MLS observations. However, the geopotential height simulated by CESM2-WACCM is relatively higher than that in the NCEP reanalysis. The simulations of CO concentration in the other 5 models show a relatively larger bias compared to the observations. The geopotential height fields simulated by CESM2-WACCM-FV2 are similar to those in the NCEP reanalysis. From the perspective of the chemical and dynamical fields, CESM2-WACCM depicts the better distribution of CO within the ASMA.

The vertical structure of the ASMA provides solid evidence of CO transport to the tropopause. The upward motions in the troposphere lead to enhanced CO concentration over this region. Taking all the features into consideration, CESM2-WACCM simulation is the closest to observations from the perspectives of CO concentration, CO vertical distribution and vertical motions.

It is found that the CO mixing ratios simulated by the six CMIP6 models all exhibit an interannual variation. However, the south–north oscillation, the east–west oscillation, and interannual variation features shown in these simulations are different from that in MLS. That is to say, these CMIP6 models cannot reproduce the interannual variation of the CO mixing ratio over the ASM region.

GFDL-ESM4 simulation has the smallest RMSE and good spatial correlation (above 0.9) compared to MLS.

Based on the simulations of CESM2-WACCM and GFDL-ESM4, future projections of JJA mean CO mixing ratio at 100 hPa over the ASM region are investigated under a series of scenarios. Under the SSP3-7.0 and SSP5-8.5 scenarios the CO mixing ratio at the end of the century will increase. It will increase to 100 ppbv, which is about 2 times larger than the 2005–2014 mean value under SSP3-7.0 scenario. The CO mixing ratio will decrease under the SSP1-1.9, SSP1-2.6 and SSP2-4.5 scenarios. After 2050, the CO mixing ratios under the SSP 2-4.5 and SSP 5-8.5 will change in different directions. The SSP3 follows the unscientific consumption of fossil fuels and the rapid expansion of the population, which will lead to continuously increasing pollutant emission.

Collectively, the six CMIP6 models offer distinct advantages and disadvantages in terms of the ASMA simulation and related regional and temporal dispersion of CO. CESM2-WACCM model shows the best performance. This paper only uses a single chemical variable CO as the tracer to evaluate the simulations. Note that the MLS dataset used in the present study is of quality level 3, which is not a dataset of high accuracy. Besides, the MLS data is not applicable near the surface below 200 hPa, which means comparisons within this region are a loss. In the future work, we can add a variety of stratospheric and tropospheric tracers to the research and use more high-quality observations to compare with model outputs. In addition, reliable low-altitude observations could be added as a supplement.

**Author Contributions:** Conceptualization, J.L.; methodology, Z.L. and J.L.; software, Z.L.; validation, Z.L., T.L. and F.L.; formal analysis, Z.L., F.Z. and T.L.; investigation, Z.L., F.Z. and T.L.; data curation, Z.L., F.Z. and F.L.; writing—original draft preparation, Z.L., F.Z., T.L., F.L. and J.L.;visualization, Z.L. and F.Z.; project administration, J.L.; funding acquisition, J.L. All authors have read and agreed to the published version of the manuscript.

**Funding:** This work was supported by the strategic priority research program of the Chinese Academy of Sciences (Grant No. XDA17010106), the National Natural Science Foundation of China (Grant No. 42075060), Science and Technology Plan of Gansu Province (20JR10RA626).

**Institutional Review Board Statement:** Not applicable.

**Informed Consent Statement:** Not applicable.

**Data Availability Statement:** The MLS satellite data applied in this work are available at https://acdisc.gesdisc.eosdis.nasa.gov/data/Aura_MLS_Level3/ML3MBCO.004/2005/, accessed on 17 November 2022. The NCEP data are obtained from https://psl.noaa.gov/data/gridded/data.ncep.reanalysis.html, accessed on 17 November 2022. The CMIP6 models can be downloaded from https://esgf-node.llnl.gov/search/cmip6/, accessed on 17 November 2022.

**Acknowledgments:** We thank the scientific teams at the National Aeronautics and Space Administration (NASA) for providing MLS satellite data. We also thank the scientific teams at National Oceanic and Atmospheric Administration (NOAA) for providing the NCEP reanalysis data. We acknowledge the World Climate Research Programme to coordinate and promote CMIP6 through its Working Group on Coupled Modelling. We thank various climate modelling groups for producing and making available their model outputs. We appreciate the Earth System Grid Federation (ESGF) for archiving the data and providing public access and the multiple funding agencies that support CMIP6 and ESGF.

**Conflicts of Interest:** The authors declare no conflict of interest.

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
