# Peer review of "Spatiotemporal Distribution of CO in the UTLS Region in the Asian Summer Monsoon Season: Analysis of MLS Observations and CMIP6 Simulations"

_remotesensing, doi:10.3390/rs15020367_

Round 1

Reviewer 1 Report

Please find the comments from the attached file

Reviewer 2 Report

This paper presents the spatiotemporal distribution of CO in relation to the Asian summer monsoon anticyclone, using the MLS observation and the outputs of the CMIP6 models. Overall, this manuscript is well organized but has some issues with the discussion of the observation and model results. For these reasons, I recommend major revisions before publication.

Major comments:

(1) L175 and L298: Both the “dynamical and chemical fields” is not compared with MLS observations. The comparison is only for the CO concentration at 100 hPa within the ASMA.

(2) L206: The result of MLS observation is not available below 200 hPa. Similar results to CESM2-WACCM above 200 hPa are seen in EC-Earth3-AerChem and MRI-ESM2-0.

(3) L220-223 and L314-315: The coarser horizontal resolution of MLS is related to the smoothness of the longitudinal distribution. However, the phase shift discussed in this sentence is not explained by the resolution. The authors will check the transport process of models carefully.

(4) L236-242 and L318-320: The inter-annual variation of CO distribution is not well simulated with the historical experiments that include the historical surface emission with the free run of the atmospheric model. The hindcast experiment will be needed to simulate the south-north oscillation.

(5) The Summary is lengthy, and some part is duplicated from the results.

Minor comments:

(6) L41: The reference [6] is focused on the emission inventory and did not include the ASM. Thus, the reference [6] will be moved to the previous sentence.

(7) The emission inventory of CO will be described in Section 2.3, since the spatiotemporal variation of CO emission is important to the concentration of the model. Did all models use the same emission inventories?

(8) L149: The “potential temperature” is not proper in this context.
